# Unmet need for hypercholesterolemia care in 35 low- and middle-income countries: A cross-sectional study of nationally representative surveys

Maja E. Marcus[1]*, Cara Ebert[2], Pascal Geldsetzer[3,4], Michaela Theilmann[4], Brice Wilfried Bicaba[5], Glennis Andall-Brereton[6], Pascal Bovet[7,8], Farshad Farzadfar[9], Mongal Singh Gurung[10], Corine Houehanou[11], Mohammad-Reza Malekpour[9], Joao S. Martins[12], Sahar Saeedi Moghaddam[13], Esmaeil Mohammadi[9], Bolormaa Norov[14], Sarah Quesnel-Crooks[6], Roy Wong-McClure[15], Justine I. Davies[16,17,18], Mark A. Hlatky[19], Rifat Atun[20‡], Till W. Bärnighausen[4,20,21‡], Lindsay M. Jaacks[22,23‡], Jennifer Manne-Goehler[20,24‡], Sebastian Vollmer[1‡]

1 Department of Economics & Centre for Modern Indian Studies, University of Goettingen, Göttingen, Germany, 2 RWI-Leibniz Institute for Economic Research, Essen (Berlin Office), Germany, 3 Division of Primary Care and Population Health, Department of Medicine, Stanford University, Stanford, California, United States of America, 4 Heidelberg Institute of Global Health, Heidelberg University and University Hospital, Heidelberg, Germany, 5 Ministry of Health, Ouagadougou, Burkina Faso, 6 Caribbean Public Health Agency, Port of Spain, Trinidad and Tobago, 7 Ministry of Health, Victoria, Seychelles, 8 University Centre for General Medicine and Public Health (Unisanté), Lausanne, Switzerland, 9 Non-Communicable Diseases Research Center, Endocrinology and Metabolism Population Sciences Institute, Tehran University of Medical Sciences, Tehran, Iran, 10 Health Research and Epidemiology Unit, Ministry of Health, Thimphu, Bhutan, 11 Laboratory of Epidemiology of Chronic and Neurological Diseases, Faculty of Health Sciences, University of Abomey-Calavi, Abomey-Calavi, Benin, 12 Faculty of Medicine and Health Sciences, Universidade Nacional Timor Lorosa'e, Dili, Timor-Leste, 13 Endocrinology and Metabolism Research Center, Endocrinology and Metabolism Clinical Sciences Institute, Tehran University of Medical Sciences, Tehran, Iran, 14 National Center for Public Health, Ulaanbaatar, Mongolia, 15 Office of Epidemiology and Surveillance, Caja Costarricense de Seguro Social, San Jose, Costa Rica, 16 Institute for Applied Health Sciences, University of Birmingham, Birmingham, United Kingdom, 17 MRC/Wits Rural Public Health and Health Transitions Research Unit, School of Public Health, University of Witwatersrand, Johannesburg, South Africa, 18 Centre for Global Surgery, Department of Global Health, Stellenbosch University, Cape Town, South Africa, 19 Department of Medicine, Stanford University, Stanford, California, United States of America, 20 Department of Global Health and Population, Harvard T.H. Chan School of Public Health, Boston, Massachusetts, United States of America, 21 Africa Health Research Institute, Somkhele, South Africa, 22 Global Academy of Agriculture and Food Security, University of Edinburgh, Edinburgh, United Kingdom, 23 Public Health Foundation of India, New Delhi, Delhi NCR, India, 24 Division of Infectious Diseases, Massachusetts General Hospital, Harvard Medical School, Boston, Massachusetts, United States of America

‡ These authors are joint senior authors on this work.
* maja-emilia.marcus@uni.goettingen.de

**Data Availability Statement:** Data included in this study are publicly available for 25 of the 35

## Abstract

### Background

As the prevalence of hypercholesterolemia is increasing in low- and middle-income countries (LMICs), detailed evidence is urgently needed to guide the response of health systems to this epidemic. This study sought to quantify unmet need for hypercholesterolemia care among adults in 35 LMICs.

countries. Microdata can be downloaded (upon free registration) at the following links: Bangladesh: https://dhsprogram.com/data/dataset/Bangladesh_Standard-DHS_2011.cfm?flag=0 Chile: https://www.minsal.cl/estudios_encuestas_salud/ The country surveys included in this analysis that are publicly available through the STEPS Microdata repository (https://extranet.who.int/ncdsmicrodata/index.php/catalog/STEPS) are: Algeria 2016-2017, Azerbaijan 2017, Belarus 2015, Benin 2014, Bhutan 2014, Botswana 2014, Eswatini 2014, Guyana 2016*, Iraq 2015, Kiribati 2015, Kyrgyzstan 2013, Lebanon 2017, Moldova 2013, Mongolia 2013, Morocco 2017, Myanmar 2014, Solomon Islands 2015, Sudan 2016, Tajikistan 2016-2017, Timor-Leste 2014, Tuvalu 2015, Vietnam 2015, Zambia 2017. For the remaining countries, please contact ghp@hsph.harvard.edu. *For Guyana and St. Vincent and the Grenadines, which are member countries of the Caribbean Public Health Agency (CARPHA): Data were originally shared through a Data Use Agreement signed with the Executive Director of CARPHA and the agreement of The Ministries of Health of St. Vincent and the Grenadines and Guyana. The Chief Medical Officer of the St. Vincent and the Grenadine's Ministry of Health (Dr. Simone Keizer-Beache) can be contacted, if necessary. Of note, after our Data Use Agreement was signed, microdata for the Guyana 2016 STEPS survey also became publicly available on the STEPS Microdata repository.

**Funding:** JMG was supported by Grant Number T32 AI007433 from the National Institute of Allergy and Infectious Diseases. The contents of this research are solely the responsibility of the authors and do not necessarily represent the official views of the NIH. PG was supported by the National Center for Advancing Translational Sciences of the National Institutes of Health under Award Number KL2TR003143. The funders had no role in study design, data collection and analysis, decision to publish, or preparation of the manuscript.

**Competing interests:** The authors have declared that no competing interests exist.

**Abbreviations:** AHA/ACC, American College of Cardiology/American Heart Association Task Force on Clinical Practice Guidelines; ATP III, Adults Treatment Panel III; BMI, body mass index; CVD, cardiovascular disease; DALY, disability-adjusted life year; EAS, European Atherosclerosis Society; ESC, European Society of Cardiology; LDL-C, low-density lipoprotein cholesterol; LMICs, low- and middle-income countries; NCD, noncommunicable disease; POC, point-of-care; RR, risk ratio; STROBE, Strengthening the Reporting of Observational Studies in Epidemiology; TC, total

## Methods and findings

We pooled individual-level data from 129,040 respondents aged 15 years and older from 35 nationally representative surveys conducted between 2009 and 2018. Hypercholesterolemia care was quantified using cascade of care analyses in the pooled sample and by region, country income group, and country. Hypercholesterolemia was defined as (i) total cholesterol (TC) $\geq$240 mg/dL or self-reported lipid-lowering medication use and, alternatively, as (ii) low-density lipoprotein cholesterol (LDL-C) $\geq$160 mg/dL or self-reported lipid-lowering medication use. Stages of the care cascade for hypercholesterolemia were defined as follows: screened (prior to the survey), aware of diagnosis, treated (lifestyle advice and/or medication), and controlled (TC <200 mg/dL or LDL-C <130 mg/dL). We further estimated how age, sex, education, body mass index (BMI), current smoking, having diabetes, and having hypertension are associated with cascade progression using modified Poisson regression models with survey fixed effects.

High TC prevalence was 7.1% (95% CI: 6.8% to 7.4%), and high LDL-C prevalence was 7.5% (95% CI: 7.1% to 7.9%). The cascade analysis showed that 43% (95% CI: 40% to 45%) of study participants with high TC and 47% (95% CI: 44% to 50%) with high LDL-C ever had their cholesterol measured prior to the survey. About 31% (95% CI: 29% to 33%) and 36% (95% CI: 33% to 38%) were aware of their diagnosis; 29% (95% CI: 28% to 31%) and 33% (95% CI: 31% to 36%) were treated; 7% (95% CI: 6% to 9%) and 19% (95% CI: 18% to 21%) were controlled. We found substantial heterogeneity in cascade performance across countries and higher performances in upper-middle-income countries and the Eastern Mediterranean, Europe, and Americas. Lipid screening was significantly associated with older age, female sex, higher education, higher BMI, comorbid diagnosis of diabetes, and comorbid diagnosis of hypertension. Awareness of diagnosis was significantly associated with older age, higher BMI, comorbid diagnosis of diabetes, and comorbid diagnosis of hypertension. Lastly, treatment of hypercholesterolemia was significantly associated with comorbid hypertension and diabetes, and control of lipid measures with comorbid diabetes. The main limitations of this study are a potential recall bias in self-reported information on received health services as well as diminished comparability due to varying survey years and varying lipid guideline application across country and clinical settings.

## Conclusions

Cascade performance was poor across all stages, indicating large unmet need for hypercholesterolemia care in this sample of LMICs—calling for greater policy and research attention toward this cardiovascular disease (CVD) risk factor and highlighting opportunities for improved prevention of CVD.

## Author summary

### Why was this study done?

- The prevalence of hypercholesterolemia is increasing in low- and middle-income countries (LMICs).

cholesterol; WHO PEN, WHO Package of Essential Noncommunicable Disease Interventions for Primary Health Care in Low-Resource Settings; WHO, World Health Organization.

- Evidence on how well health systems address this rising hypercholesterolemia burden is limited. Nationally representative studies analyzing care at the individual level across a larger number of LMICs are largely missing.

## What did the researchers do and find?

- **We analyzed access to hypercholesterolemia care using pooled data from 35 nationally representative, individual-level surveys from LMICs.**

- **We found** a prevalence of high total cholesterol (TC) of 7.1% (95% CI: 6.8% to 7.4%) and a high low-density lipoprotein cholesterol (LDL-C) prevalence of 7.5% (95% CI: 7.1% to 7.9%) in this set of countries.

- **Using a cascade of care approach, we found that** 43% (95% CI: 40% to 45%) of individuals with high TC and 47% (95% CI: 44% to 50%) with high LDL-C ever had their cholesterol measured prior to the survey. About 31% (95% CI: 29% to 33%) and 36% (95% CI: 33% to 38%) were aware of their diagnosis; 29% (95% CI: 28% to 31%) and 33% (95% CI: 31% to 36%) were treated; 7% (95% CI: 6% to 9%) and 19% (95% CI: 18% to 21%) were controlled.

- **Using modified Poisson regression models, we found that access to care was significantly associated with a range of sociodemographic characteristics, such as high education and old age, as well as with the presentation of other cardiovascular disease (CVD) risk factors, such as comorbid diabetes or hypertension and a high body mass index.**

## What do these findings mean?

- We found large unmet need for hypercholesterolemia care in this sample of LMICs.

- This calls for greater policy and research attention toward this CVD risk factor.

- High-performing countries, such as Sri Lanka, Costa Rica, Iran, and Morocco, may highlight policy opportunities for improved prevention of CVD.

- The main limitations of this study are a potential recall bias in self-reported information on received health services as well as diminished comparability due to varying survey years and varying lipid guideline application across country and clinical settings.

## Introduction

Cardiovascular disease (CVD) is already the leading cause of death in low- and middle-income-countries (LMICs) and is projected to increase rapidly in the coming decades [1,2]. Hypercholesterolemia—defined as abnormal levels of blood lipids, such as high fasting total cholesterol (TC)—is the second leading physiological risk factor for CVD after high blood pressure [3,4]. High cholesterol was estimated to cause 3.5 million deaths and 81.4 million

disability-adjusted life years (DALYs) in LMICs in 2019 [3]. Importantly, the disease burden caused by hypercholesterolemia is eminently preventable with lifestyle modification and low-cost, off-patent medications [4–7]. The fact that a high burden persists suggests that many health systems in LMICs are still ill-equipped to address this important condition.

Despite the importance of rigorous evidence to guide health policy and improve healthcare delivery, the current empirical evidence remains weak and offers only a limited understanding of the state of care for hypercholesterolemia in LMICs [8,9]. Research is mainly confined to single countries, often based on a subnational level with a focus on specific subpopulations, or to single healthcare indicators, such as access to essential medicines [8,10–13]. To our knowledge, nationally representative studies analyzing broader health system performance at the individual level across a larger number of LMICs have been altogether absent.

Our analysis aims to address this dearth of evidence by identifying the unmet need for hypercholesterolemia care using a pooled dataset of nationally representative, population-based surveys that includes 129,040 individuals from 35 LMICs. We assess the unmet need for care by applying the cascade of care approach, a quantitative depiction of the screening, diagnosis, treatment, and control stages within the care system of the affected population groups. This methodology has been widely used to monitor care responses to the HIV epidemic and is increasingly applied to examine the management of chronic diseases, such as diabetes or hypertension [14–17]. We estimate the cascade of care for individuals, separately for high TC and high low-density lipoprotein cholesterol (LDL-C), (i) in a pooled sample across all 35 LMICs and (ii) disaggregated at the World Health Organization's (WHO) epidemiological subregion [18], World Bank country income classification [19], and country level. We then estimate the associations between individual-level characteristics and cascade completion—yielding insights into the overall unmet need for care as well as into potentially underserved subpopulations in this group of LMICs.

## Methods

### Ethics

This study received a determination of "not human subjects research" by the institutional review board of the Harvard T.H. Chan School of Public Health.

### Data sources

The included datasets were obtained through a systematic request approach. We first targeted surveys following the WHO's Stepwise Approach to Surveillance of Noncommunicable Disease (NCD) Risk Factors (STEPS). We identified responsible contacts for each survey via the WHO STEPS website, expert contacts, a web search, and the WHO NCD Microdata repository [20]. Inclusion criteria were as follows: surveys had to be conducted during or after 2008; had to come from an upper-middle, lower-middle, or low-income country per World Bank definition during the survey year [19]; be nationally representative with a response rate of over 50%; have data available at the individual level; include biomarkers for hypercholesterolemia (TC or LDL-C); and include questions that assess the access to health services for diagnosis, preventive counseling, and treatment of hypercholesterolemia. Whenever STEPS surveys were not available, we searched for complementing data meeting the inclusion criteria. A detailed protocol and outcome of the search process is provided in S1 Text.

This process yielded 32 STEPS surveys from 2010 to 2018 to be included in our analysis: Algeria, Azerbaijan, Bangladesh, Belarus, Benin, Bhutan, Botswana, Burkina Faso, Costa Rica, Ecuador, Eswatini, Guyana, Iran, Iraq, Kiribati, Kyrgyzstan, Lebanon, Moldova, Mongolia,

Morocco, Myanmar, Solomon Islands, Sri Lanka, St. Vincent and the Grenadines, Sudan, Tajikistan, Timor-Leste, Tokelau, Tonga, Tuvalu, Vietnam, and Zambia.

Supplementary to these, we added the 2009/10 National Health Survey from Chile, the 2013 National Survey of Noncommunicable Diseases from Seychelles, and the 2017 HYBRID Survey from the Marshall Islands. All surveys used multistage cluster random sampling to select participants. Details on the sampling strategies can be found in S2 Text.

## Cascade construction

The cascade-of-care methodology first requires the identification of individuals with hypercholesterolemia to serve as the overall sample. Our definition of hypercholesterolemia is contingent upon a collected biomarker sample and self-reported medication use. We used 2 lipid biomarkers to establish a set of hypercholesterolemia definitions—TC and LDL-C. TC is significantly and positively associated with ischemic heart disease mortality as well as other CVDs and is the most commonly measured lipid biomarker in the LMIC literature [4,21]. LDL-C is the primary target for cholesterol-lowering therapy according to the Adults Treatment Panel III (ATP III) guidelines of the National Cholesterol Education Program and therefore holds particular relevance for the analysis of unmet need for care [4,22]. The biomarker cutoffs for classifying hypercholesterolemia are based on the ATP III guidelines, which are frequently used in the literature [23–26]. The ATP III provides 3 classifications of TC and 5 classifications of LDL. The 3 TC classifications are as follows: (i) TC <200 mg/dL is classified as "desirable"; (ii) $200 \leq TC \leq 239$ is classified as "borderline high"; and (iii) TC $\geq$240 is classified as "high". The 5 LDL classifications are as follows: (i) LDL-C <100 is classified as "optimal"; (ii) $100 \leq LDL\text{-}C \leq 129$ is classified as "near or above optimal"; (iii) $130 \leq LDL\text{-}C \leq 159$ is classified as "borderline high"; (iv) $160 \leq LDL\text{-}C \leq 189$ is classified as "high"; and (v) LDL-C $\geq$129 is classified as "very high" [4]. We defined hypercholesterolemia as "high" and "very high" lipid values. We opted for this classification for 2 reasons. First, "high" TC and LDL-C values have been shown to be associated with an increased lifetime risk of coronary heart disease justifying clinical therapies and necessitating care [4]. Second, treatment guidelines are usually based on CVD risk scores rather than on lipid measures alone and often vary across countries [4,22]. In order to not evaluate health systems by care standards that are in fact unapplied, we chose to be conservative in our definition of hypercholesterolemia. Thus, we defined hypercholesterolemia based on respondents with (i) a TC measurement of 240 mg/dL or higher or who were taking lipid-lowering medication and, alternatively, (ii) an LDL-C measurement of 160 mg/dL or higher or taking lipid-lowering medication. However, in supplemental analyses, we redefine hypercholesterolemia to further include "borderline high" values as well as apply a definition based on the American College of Cardiology/American Heart Association Task Force on Clinical Practice Guidelines (AHA/ACC) (AHA/ACC uses an LDL-C cutoff of 70 mg/dL as a threshold for statin therapy in adults 40 to 75 years of age with diabetes or with a 10-year atheroslerotic CVD risk of over 7.5%. In our definition, we classify everyone with an LDL-C measurement of 70 mg/dL or higher as having hypercholesterolemia) (see S1A–S1C Fig).

Bangladesh, Chile, Costa Rica, Guyana, Iran, Iraq, and Lebanon measured lipid biomarkers via blood samples sent to a laboratory. Belarus, Benin, Bhutan, Burkina Faso, Ecuador, Eswatini, Kiribati, Moldova, Morocco, Solomon Islands, Sri Lanka, St. Vincent and the Grenadines, Sudan, Timor-Leste, Tonga, Vietnam, and Zambia used the CardioCheck PA point-of-care (POC) device. Seychelles used the Konelab 30i, Mongolia the Prima Home Test, Myanmar the SD Lipido Care Analyzer, Tokelau the Accutrend GC, and Tuvalu the Accutrend Plus (see

S3 Text). For the remaining 6 countries, we could not identify whether biomarkers were measured via a laboratory or a POC machine.

In Algeria, Bangladesh, Burkina Faso, Chile, Costa Rica, Iran, Iraq, Lebanon, Mongolia, Morocco, Myanmar, Seychelles, and St. Vincent and the Grenadines, both TC and LDL-C records were collected, while the remaining countries measured only TC. We took TC records directly from the survey and derived LDL-C from TC, triglycerides, and HDL cholesterol records using the Friedewald equation [27]. Individuals without a biomarker record were excluded from the analysis (S4 Text). A sensitivity analysis that includes individuals with no biomarker measurement, for whom hypercholesterolemia is defined purely based on the self-reported medication status, can be found in S1D Fig. We further excluded observations with TC records above 300 mg/dL because, even though physiologically very high TC values may occur, POC devices are not always well equipped to reliably measure these (S4 Text) [28,29]. Supplementary analyses including TC values above 300 mg/dL can be found in S1E Fig.

In a next step, the cascade-of-care analysis requires the measurement of the sample respondents' met need for hypercholesterolemia care prior to the survey. For this, we defined the following 4 cascade stages expressing each step in the care continuum: (1) ever received a cholesterol measurement ("Lipids Measured"); (2) ever been told by a healthcare professional about one's hypercholesterolemia diagnosis ("Aware of Diagnosis"); (3) received lifestyle advice or currently taking medication for high cholesterol ("Advice or Medication"); and (4) has lipid measure in controlled ranges ("Controlled Disease"). Our definition of the last cascade stage was again based on biomarker measurements. We recognize that there usually are no clinical target ranges for cholesterol alone, and, thus, we chose to define "controlled" lipid ranges based on the ATP III guidelines' definition of "desirable", "optimal", and "near optimal" values, as was done in related literature [30–32]. Hence, according to our definition, an individual had controlled lipid values whenever TC was lower than 200 mg/dL and LDL-C was lower than 130 mg/dL [4]. Supplementary cascade analyses based on a definition that further considers "borderline high" values ($\geq$200 and <240 mg/dL TC; $\geq$130 and <160 mg/dL LDL-C) as "controlled" lipid values can be found in S1F Fig.

The cascade stages "Lipids Measured", "Aware of Diagnosis", and "Advice or Medication" were measured with self-reported interview data. Across surveys, the question phrasing of these cascade measures was almost identical as is shown in S4 Text. "Lipids Measured" refers to lipid measurements that had taken place prior to the survey. For "Advice or Medication" (3), advice refers to lifestyle advice about physical activity, body weight, fruit and vegetable intake, special diets, reduction of fat, or tobacco consumption. Medication refers to any oral treatment for high cholesterol.

## Statistical analysis

In the cascade-of-care analysis, we calculated the share of respondents that reach each consecutive cascade stage over the denominator of all individuals with hypercholesterolemia defined either as high TC or as high LDL-C. We estimated the cascades of care for the pooled sample as well as at the WHO epidemiological subregion, World Bank country income classification, and country level for both hypercholesterolemia definitions. In addition to this, we carried out a pooled cascade analysis on a restricted sample of individuals with hypercholesterolemia for whom cholesterol screening is recommended according to international guidelines. This allows an examination of health system performance in relation to adherence to approved care guidelines. We derived the screening recommendation guidelines from the WHO Package of Essential Noncommunicable (PEN) Disease Interventions for Primary Health Care in Low-Resource Settings [33]. The PEN protocol specifies that individuals exhibiting any one of the

following risk factors should be included in the routine management of CVD risk and undergo cholesterol screening: age >40 years; current smoking; waist circumference ≥90 cm in males or ≥100 cm in females; having diabetes; or having hypertension [33].

We adjusted all cascade estimations for survey sampling designs using the "svy set" command with subpopulation specifications in Stata 16.1 (StataCorp, College Station, Texas, United States), used R's ggplot2 package for the disaggregated cascade graphics, and R's gee-pack package as well as Stata 16.1 for regression estimations [34].

In addition to the cascade analyses, we estimated individual-level correlates of cascade progression. We regressed the proportion of respondents with high TC or high LDL-C that reached each cascade stage on age, sex, education, smoking, body mass index (BMI), diabetes, and hypertension status. In this, we adjusted our standard errors for clustering at the primary sampling unit level and included survey fixed effects (for mathematical equations, see S5 Text). The regression analyses were not weighted [35]. We used a modified Poisson regression model yielding risk ratios (RRs) as our main specification and supplemented our analysis with additional univariable and multivariable models and an analysis of deviance in S2 Table [36].

## Covariate measurement

Age, smoking status, and education were self-reported. Sex was recorded as observed. We calculated respondents' BMI from height and weight measurements that were taken alongside waist circumference measurements. The hypertension status was derived from blood pressure readings and the diabetes status from collected blood glucose measurements. Hypertension was defined as a systolic blood pressure of at least 140 mm Hg, diastolic blood pressure of at least 90 mm Hg, or reported use of medication for hypertension. Diabetes was defined as fasting plasma glucose of at least 7.0 mmol/l (126 mg/dl), random plasma glucose of at least 11.1 mmol/l (200 mg/dl), HbA1c of at least 6.5%, or reporting to be taking medication for diabetes. More details on the definition and measurement of these comorbidities are provided elsewhere [15,16].

## STROBE guidelines

This study is reported as per the Strengthening the Reporting of Observational Studies in Epidemiology (STROBE) guideline (see S6 Text).

## Results

### Sample characteristics

Our sample included 129,040 individuals from 35 LMICs over a 9-year period (2009 to 2018). Details on country-specific sample characteristics can be found in S1A Table. Sociodemographic characteristics of the respondents are displayed in Table 1 stratified by biomarkers. We found that 7.1% (95% CI: 6.8% to 7.4%) of individuals had high TC and 7.5% (95% CI: 7.1% to 7.9%) had high LDL-C (also see S1C Table). The mean age of the overall sample was around 40 to 41 years (SD: 14 years), whereas the mean age in those with either form of hypercholesterolemia was around 49 years (SD: 13 to 14 years). Secondary schooling or higher education was completed by 41% of those with high TC and 31% of those with high LDL-C. Around 59% to 61% of those with hypercholesterolemia were overweight or obese, and approximately 16% to 17% were current smokers. Comorbid diabetes occurred in 23% to 24% of those with hypercholesterolemia and hypertension in 49% to 52%. Around 87% to 89% of those with hypercholesterolemia exhibited at least 1 associated risk factor, indicating that cholesterol screening was recommended for them. Sample characteristics of respondents who had

**Table 1. Sociodemographic sample characteristics by hypercholesterolemia definition.**

| | TC Sample* | | | | LDL-C Sample** | | | |
| --- | --- | --- | --- | --- | --- | --- | --- | --- |
| | Overall Sample | | Sample With High TC | | Overall Sample | | Sample With High LDL-C | |
| | Number of Observations[†] | Percentage or Mean[‡] | Number of Observations[†] | Percentage or Mean[‡] | Number of Observations[†] | Percentage or Mean[‡] | Number of Observations[†] | Percentage or Mean[‡] |
| Hypercholesterolemia Prevalence*** | 128,998 | 7 | 10,737 | 100 | 58,332 | 7 | 6,315 | 100 |
| Female | 128,996 | 51 | 10,732 | 51 | 58,330 | 52 | 6,314 | 59 |
| Age(mean) | 128,998 | 40 | 10,733 | 49 | 58,332 | 41 | 6,315 | 49 |
| 15–24 y/o | 12,290 | 15 | 194 | 3 | 3,184 | 12 | 88 | 3 |
| 25–34 y/o | 29,555 | 26 | 820 | 13 | 12,882 | 26 | 512 | 14 |
| 35–44 y/o | 30,445 | 23 | 1,713 | 19 | 14,278 | 24 | 1,024 | 19 |
| 45–54 y/o | 26,964 | 18 | 2,967 | 28 | 12,824 | 19 | 1,689 | 27 |
| 55–64 y/o | 20,757 | 13 | 3,328 | 26 | 9,728 | 13 | 1,842 | 24 |
| 65+ y/o | 9,029 | 5 | 1,715 | 11 | 5,436 | 6 | 1,160 | 13 |
| Education | | | | | | | | |
| Less than primary school | 25,566 | 21 | 1,906 | 24 | 12,719 | 26 | 1,456 | 29 |
| Less than secondary school | 39,406 | 34 | 3,470 | 35 | 20,784 | 39 | 2,376 | 39 |
| Secondary completed or higher | 62,086 | 45 | 5,064 | 41 | 23,767 | 35 | 2,292 | 31 |
| BMI | | | | | | | | |
| Normal | 53,969 | 52 | 2,750 | 38 | 22,596 | 48 | 1,584 | 36 |
| Underweight | 8,323 | 10 | 231 | 3 | 3,538 | 9 | 113 | 3 |
| Overweight | 36,438 | 25 | 3,790 | 37 | 18,591 | 28 | 2,385 | 38 |
| Obese | 28,024 | 13 | 3,715 | 22 | 12,465 | 15 | 2,068 | 23 |
| Smoking# | 128,329 | 20 | 10,699 | 16 | 57,974 | 20 | 6,292 | 17 |
| Diabetic | 121,887 | 8 | 10,062 | 24 | 57,288 | 9 | 6,190 | 23 |
| Hypertensive | 127,755 | 27 | 10,650 | 52 | 57,766 | 27 | 6,265 | 49 |
| Screening recommended§ | 128,998 | 68 | 10,733 | 89 | 58,332 | 70 | 6,315 | 87 |
| **Total Number of Observation** | 128,998 | | 10,737 | | 58,332 | | 6,315 | |

*Includes respondents from all 32 countries with a valid TC measurement (see S4 Text); columns "Sample With High TC" restricted to respondents with high TC (defined by exceeding ATP III guideline cutoffs, i.e., TC ≥6.21 mmol/L, or respondent taking lipid medication).

**Includes respondents from Algeria, Bangladesh, Burkina Faso, Chile, Costa Rica, Iran, Iraq, Lebanon, Mongolia, Morocco, Myanmar, Seychelles, and St. Vincent and the Grenadines with a valid LDL-C measurement (see S4 Text); columns "Sample With High LDL-C" restricted to respondents with high LDL-C (defined by exceeding ATP III guideline cutoffs, i.e., LDL-C ≥4.14 mmol/L, or respondent taking lipid medication).

***Refers to high TC in columns 1–4 and high LDL-C in columns 5–8. See S1C Table for 95% confidence intervals.

[†]Unweighted.

[‡]Values account for sampling design with survey weights rescaled by the survey's sample size such that all countries contribute to estimates according to their population size.

#Respondents that are currently smoking or were smoking within past 12 months are classified as smoking (as per WHO PEN disease interventions for primary healthcare in low-resource settings (WHO PEN) Protocol 1).

§According to the PEN protocol, screening is recommended whenever the respondent exhibits at least one of the following risk factors: age >40; smoking; diabetic; hypertensive; waist circumference > = 90 in males; waist circumference > = 100 in females.

ATP III, Adults Treatment Panel III; BMI, body mass index; LDL-C, low-density lipoprotein cholesterol; TC, total cholesterol; WHO PEN, World Health Organization package of essential noncommunicable disease interventions for primary healthcare in low-resource settings.

TC and LDL-C measures in normal ranges and who reported not taking lipid-lowering medication can be found in S1B Table.

## Pooled cascades of care

The cascades of care for the pooled country sample are displayed in Fig 1A and 1B. Only 43% (95% CI: 40% to 45%) of those with high TC and 47% (95% CI: 44% to 50%) with high LDL-C had had their blood lipids measured prior to the survey. In those with high TC (Fig 1A), 31% (95% CI: 29% to 33%) were diagnosed, and 29% (95% CI: 28% to 31%) were treated. Only 7% (95% CI: 6% to 9%) of individuals with high TC achieved control. Of those with high LDL-C (Fig 1B), less than half were diagnosed (36%, 95% CI: 33% to 38%), 33% (95% CI: 31% to 36%) were treated, and 19% (95% CI: 18% to 21%) achieved control.

Fig 1C and 1D display cascade results for individuals meeting PEN criteria for lipid screening. Cascade performance was found to be similar compared to the previous analyses: Only 45% of respondents with high TC and for whom screening was recommended according to PEN guidelines had undergone a cholesterol measurement.

Furthermore, S1 Fig shows a range of supplementary analyses. The cascade of care based on a definition that also classifies "borderline high" TC as hypercholesterolemia shows by design a substantially poorer performance. Similarly, cascade performance is markedly worse when basing the hypercholesterolemia definition on cutoffs from the AHA/ACC guidelines. The cascades of care for high TC restricted to the countries that collected both TC and LDL-C records mirrored those for high LDL-C care. The cascade of care restricted to respondents aged 40 or older mirrors the cascade results for individuals meeting PEN criteria for lipid screening. Finally, neither estimates including TC records over 300 mg/dL nor those using a more inclusive definition of controlled lipid values show substantial differences in cascade performance in comparison to the cascade of care presented in Fig 1.

## Disaggregated cascade of care

Fig 2 displays the cascades of care disaggregated by WHO epidemiological subregion, World Bank country income class, and country (for results in table format, see S2 Table). The Americas and Eastern Mediterranean and Europe regions achieved comparatively high cascade of care levels: 66% (95% CI: 61% to 71%) of individuals with hypercholesterolemia in the Americas and 52% (95% CI: 49% to 55%) of those in the Eastern Mediterranean and Europe regions have ever had their cholesterol measured. Examining the same cascade stage for Africa and Southeast Asia and Western Pacific, we found shares of 29% (95% CI: 21% to 40%) and 34% (95% CI: 30% to 38%), respectively. As the cascade progresses, all regions converge to under 15% at the control stage. We found substantial heterogeneity across countries. Iran displayed the best cascade performance—89% (95% CI: 88% to 91%) of individuals with hypercholesterolemia have had their cholesterol measured prior to the survey, and around 57% (95% CI: 54% to 60%) were still retained at the control stage. Other high-performing countries included Costa Rica, Belarus, Ecuador, Morocco, and Sri Lanka (see S2D Table). Benin, Bhutan, Eswatini, Kiribati, Myanmar, Solomon Islands, and Zambia exhibited the greatest unmet need for care. In each case, fewer than 20% of those with hypercholesterolemia ever had their cholesterol measured leaving the consecutive cascade stages at very low levels. Achieved levels of control were low almost in all of the 32 countries, with less than 10% in 26 countries. Next to Iran, only Morocco achieved comparably high levels of control, where 49% (95% CI: 41% to 58%) of those with high TC reached the last cascade stage. Cascade performance was found to be consistently higher in upper-middle-income countries.

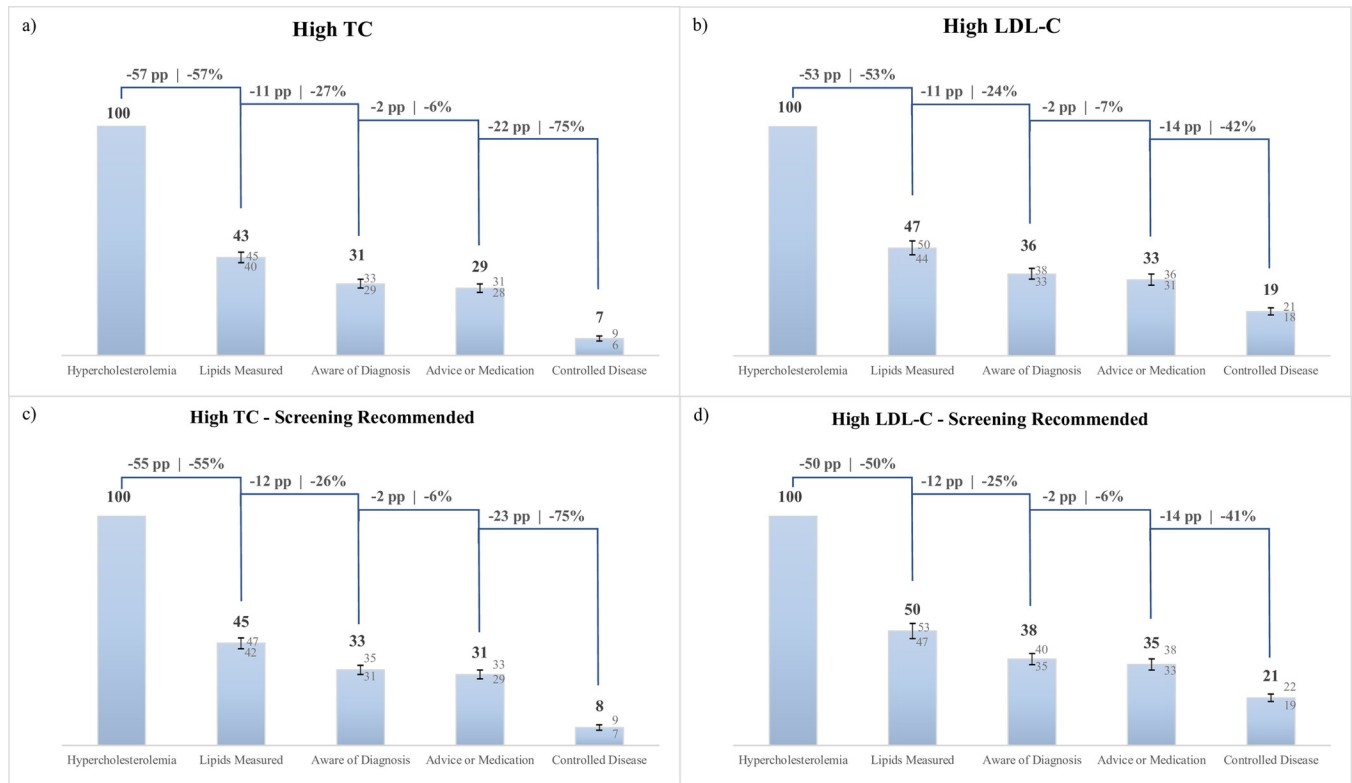

**Fig 1. Cascades of care by biomarker.** Bars represent point estimates; numeric form can be viewed above bars. Whiskers represent 95% confidence intervals; numeric form of upper and lower bounds can be viewed above and below whiskers. On top, the absolute percentage point drops of each cascade step are shown on the left-hand side and the relative percentage drop on the right-hand side. Note: All calculations incorporate PSUs and strata to account for the different survey designs of included countries, as well as use sampling weights rescaled such that all countries contribute equally. Percentage and percentage point drops are calculated with unrounded point estimates. Hypercholesterolemia refers to all respondents that are classified as having high TC, i.e., TC ≥240 mg/dL, or high LDL-C, i.e., LDL-C ≥160 mg/dL, or a self-reported medication status. Lipids Measured refers to the percentage share of all respondents with hypercholesterolemia (classified based on respective biomarker) that have ever had their lipid status measured prior to the survey as per self-reported information. Accordingly, Aware of Diagnosis refers to the percentage share of all participants with hypercholesterolemia that have (self-reportedly) ever been diagnosed by a medical professional with hypercholesterolemia, whereas Advice or Medication refers to those that have received medication or lifestyle advice for their disease. Controlled Disease considers those respondents that have TC and LDL-C values within the range considered normal by ATP III guidelines. Panel (a) only considers TC and the self-reported medication status in the classification of having hypercholesterolemia. Panel (b) only considers LDL-C and the self-reported medication status in the classification of having hypercholesterolemia. Included are all countries that measured LDL-C, namely, Algeria, Bangladesh, Burkina Faso, Chile, Costa Rica, Iran, Iraq, Mongolia, Morocco, Myanmar, Seychelles, and St. Vincent and the Grenadines. Panel (c) again considers TC and the self-reported medication status in the classification of hypercholesterolemia. It further restricts the sample to those respondents with hypercholesterolemia for which screening is recommended based on the exhibition of at least one of the following risk factors: age >40; current smoking; having diabetes; having hypertension; waist circumference ≥90 in males and ≥100 in females. Panel (d) again considers LDL-C and the self-reported medication status in the classification of having hypercholesterolemia. It further restricts the sample again to those respondents with hypercholesterolemia for which screening is recommended (as in Panel c). Included are all countries that measured LDL-C, namely, Algeria, Bangladesh, Burkina Faso, Chile, Costa Rica, Iran, Iraq, Lebanon, Mongolia, Morocco, Myanmar, Seychelles, and St. Vincent and the Grenadines. ATP III, Adults Treatment Panel III; LDL-C, low-density lipoprotein cholesterol; PSU, primary sampling unit; TC, total cholesterol.

## Individual-level characteristics and cascade progression

In estimating the association between individual-level characteristics and cascade progression, we found a significant age gradient for reaching the first and second cascade stages—for instance, over-65-year-olds were 2.09 times (95% CI: 1.67 to 2.60; *p*-value: <0.001) more likely to have had their cholesterol measured in comparison to the youngest age group (see Table 2). The age gradient disappeared in the treatment cascade stage and was found to be insignificant in the control stage. We further observed that women were significantly more likely to have

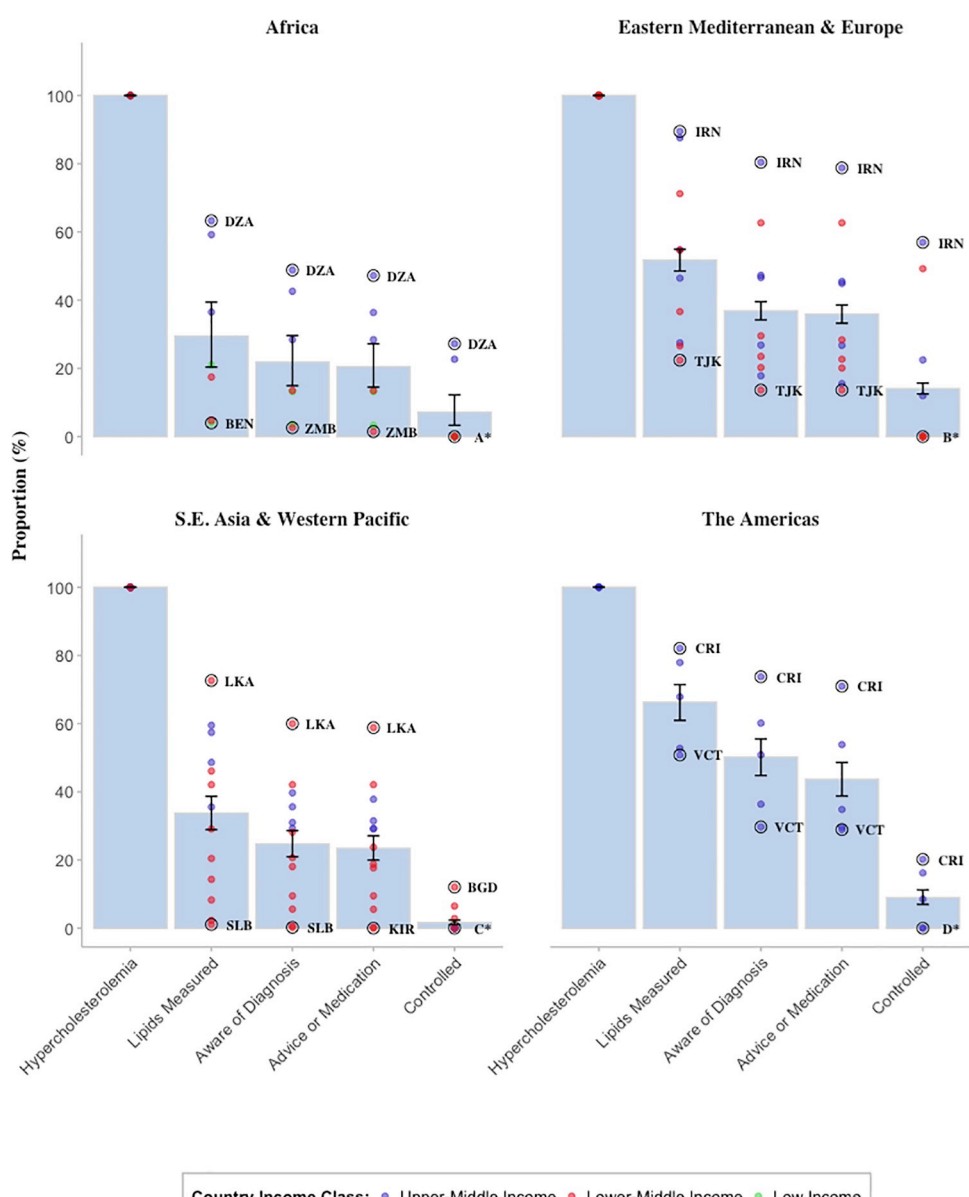

**Fig 2. Cascade of care for high TC by WHO epidemiological subregion and World Bank GDP income classification.** Bars represent pooled region point estimates. Whiskers represent pooled region 95% confidence intervals. Dots represent country point estimates; dots are color coded by GDP income classification; highest and lowest performing country of each region is indicated by country abbreviation. Note: Several countries have point estimates of zero at the control stage, in which case they were abbreviated by the letters A\*, B\*, and C\*. A\*: Benin, Botswana, Burkina Faso, Eswatini, and Zambia. B\*: Azerbaijan, Belarus, Kyrgyzstan, Moldova, Sudan, and Tajikistan. C\*: Bhutan, Kiribati, Marshall Islands, Solomon Islands, Sri Lanka, Timor-Leste, Tokelau, Tonga, Tuvalu, and Vietnam. D\*: Ecuador and Guyana. The country abbreviations follow the ISO 3166-1Alpha-3 codes: BEN, Benin; BGD, Bangladesh; CRI, Costa Rica; DZA, Algeria; GUY, Guyana; IRN, Iran; KIR, Kiribati; LKA, Sri Lanka; SLB, Solomon Islands; TJK, Tajikistan; VCT, St. Vincent and the Grenadines; ZMB, Zambia. Other abbreviations: S.E. Asia, Southeast Asia; TC, total cholesterol. For more details, see note.

been screened than men (RR: 1.06; 95% CI: 1.03 to 1.10; *p*-value: <0.001) but less likely reach the control stage (RR: 0.92; 95% CI: 0.86 to 0.98; *p*-value: 0.007). Individuals with secondary education or higher were also significantly more likely to have been screened compared to

**Table 2. Correlates of cascade progression.**

| | Measured | | | Diagnosed | | | Treated | | | Controlled | | |
|---|---|---|---|---|---|---|---|---|---|---|---|---|
| | RR | | p | RR | | p | RR | | p | RR | | p |
| **Age** | | | | | | | | | | | | |
| 15–24 years | REF | | | REF | | | REF | | | REF | | |
| 25–34 years | 1.17 | [0.93,1.48] | 0.18 | 1.12 | [0.85,1.49] | 0.41 | 0.91 | [0.82,1.01] | 0.07 | 1.19 | [0.62,2.29] | 0.61 |
| 35–44 years | 1.63 | [1.30,2.03] | <0.001 | 1.31 | [1.01,1.71] | 0.04 | 0.95 | [0.87,1.03] | 0.22 | 1.34 | [0.72,2.49] | 0.35 |
| 45–54 years | 1.86 | [1.49,2.32] | <0.001 | 1.41 | [1.09,1.83] | 0.009 | 0.96 | [0.89,1.04] | 0.33 | 1.39 | [0.75,2.57] | 0.30 |
| 55–64 years | 2.04 | [1.64,2.55] | <0.001 | 1.46 | [1.13,1.89] | 0.004 | 0.97 | [0.90,1.06] | 0.54 | 1.43 | [0.78,2.65] | 0.25 |
| 65 or older | 2.09 | [1.67,2.60] | <0.001 | 1.43 | [1.10,1.85] | 0.007 | 0.99 | [0.91,1.07] | 0.73 | 1.61 | [0.87,2.98] | 0.13 |
| **Sex** | | | | | | | | | | | | |
| Male | REF | | | REF | | | REF | | | REF | | |
| Female | 1.06 | [1.03,1.10] | <0.001 | 1.01 | [0.98,1.04] | 0.53 | 0.99 | [0.97,1.01] | 0.22 | 0.92 | [0.86,0.98] | 0.007 |
| **Education** | | | | | | | | | | | | |
| Less than primary school | REF | | | REF | | | REF | | | REF | | |
| Less than secondary school | 1.06 | [1.02,1.10] | 0.004 | 1.02 | [0.98,1.05] | 0.35 | 1.01 | [0.99,1.02] | 0.41 | 1.03 | [0.96,1.11] | 0.45 |
| Secondary school completed or higher | 1.25 | [1.20,1.30] | <0.001 | 1.01 | [0.97,1.05] | 0.52 | 1.00 | [0.98,1.02] | 0.97 | 1.01 | [0.93,1.10] | 0.74 |
| **Smoking** | | | | | | | | | | | | |
| Past or Never | REF | | | REF | | | REF | | | REF | | |
| Current | 0.96 | [0.92,1.00] | 0.07 | 0.96 | [0.92,1.01] | 0.13 | 0.97 | [0.94,1.00] | 0.03 | 1.02 | [0.93,1.12] | 0.71 |
| **BMI** | | | | | | | | | | | | |
| Normal | REF | | | REF | | | REF | | | REF | | |
| Underweight | 0.74 | [0.62,0.88] | <0.001 | 1.01 | [0.87,1.17] | 0.89 | 0.98 | [0.91,1.06] | 0.65 | 1.16 | [0.90,1.50] | 0.26 |
| Overweight | 1.08 | [1.04,1.12] | <0.001 | 1.08 | [1.04,1.12] | <0.001 | 0.99 | [0.97,1.01] | 0.20 | 1.03 | [0.96,1.12] | 0.39 |
| Obese | 1.15 | [1.11,1.20] | <0.001 | 1.08 | [1.04,1.12] | <0.001 | 0.99 | [0.97,1.01] | 0.45 | 1.01 | [0.93,1.09] | 0.86 |
| Diabetes | 1.19 | [1.15,1.22] | <0.001 | 1.10 | [1.07,1.13] | <0.001 | 1.02 | [1.01,1.04] | <0.001 | 1.21 | [1.14,1.28] | <0.001 |
| Hypertension | 1.15 | [1.12,1.19] | <0.001 | 1.09 | [1.05,1.13] | <0.001 | 1.04 | [1.02,1.06] | <0.001 | 1.04 | [0.98,1.11] | 0.18 |
| N | 10,575 | | | 6,073 | | | 4,601 | | | 4,283 | | |

Multivariable modified Poisson regression models with robust error structure, clustering at PSU, including binary country variables (survey-level "fixed effects"), and "Lipids Measured," "Aware of Diagnosis," "Advice or Medication," and "Controlled Disease" as dependent variables. Each cascade stage estimation is conditioned on completion of prior cascade stages. The coefficients indicate RRs. 95% confidence intervals in brackets. The regression samples do not include Tokelau, due to information on education not being available, nor Tonga, due to unavailable blood glucose measurements. Survey fixed effect estimates can be viewed in S1K Fig. Respondents that are currently smoking or were smoking within past 12 months are classified as current smokers (as per WHO PEN disease interventions for primary healthcare in low-resource settings (WHO PEN) Protocol 1).

BMI, body mass index; PSU, primary sampling unit; REF, reference; RR, risk ratio; WHO PEN, WHO PEN, World Health Organization Package of Essential Noncommunicable Disease Interventions for primary healthcare in low-resource settings.

those who did not complete primary schooling (RR: 1.25; 95% CI: 1.20 to 1.30; p-value: <0.001) but showed no significant association with reaching other cascade stages. Being a smoker showed only weakly significant, negative associations with having undergone a lipid screening (RR: 0.96; 95% CI: 0.92 to 1.00; p-value: 0.07) and treatment (RR: 0.97; 95% CI: 0.94 to 1.00; p-value: 0.03). Individuals who were overweight or obese were significantly (p-value: <0.001) more likely to have been screened and diagnosed in comparison to individuals with a normal BMI. Moreover, having diabetes or hypertension were found to have RRs significantly (p-value: <0.001) greater than 1 for reaching the lipid measurement, diagnosis, or treatment stage. Diabetes further had a significant and positive association with having had a lipid measure in controlled ranges.

## Discussion

In a pooled sample of 129,040 individuals from 35 LMICs, we found that less than 1 out of every 3 respondents with hypercholesterolemia had been treated and less than 1 in 5 had achieved control. By using nationally representative data that combine individual-level bio-markers with self-reported health service utilization, our study shows that cascade performance, while poor overall, is characterized by large declines at the screening and control stage in particular. To our knowledge, this is a first application of the cascade of care methodology to such an extensive evaluation of the unmet need for hypercholesterolemia care, yielding novel insights into the shortcomings of health services in this geographically diverse group of countries.

The results of this study have several important policy implications for health system strengthening. We found that screening for hypercholesterolemia constitutes a major barrier to meeting care needs, as this stage was consistently found to have the largest or second largest amount of loss along the cascade of care. In the US and Europe, where cholesterol screening rates varied in ranges comparable to our study, screening appeared to be influenced by structural health system inequities and was found to be lower in disadvantaged groups—such as racial minorities or those with low education [37–39]. Our results show that in this set of LMICs education was also positively associated with screening. We estimated that individuals with secondary education or higher had a 25% higher likelihood of being screened relative to individuals with less than primary education. Potential reasons for this could be that additional schooling results in better health literacy and greater awareness of CVD risk or—as a proxy for wealth and social status—better access to the health system. In addition to sociodemographic characteristics, we also found the presence of other CVD risk factors, such as age, high BMI, or comorbid diabetes or hypertension, to be associated with screening. This suggests that health systems are—in accordance with WHO guidance—targeting high-risk individuals for screening. However, while many individuals with hypercholesterolemia who were included in this study presented with at least 1 other CVD risk factor, cascade performance did not improve overall when examining this group only. This suggests that a large proportion of high-risk individuals were still left out of screening efforts. In cases where this relates to a lack of laboratory infrastructure and equipment as well as accessibility and affordability of care, POC machines may have the potential to increase screening rates among all population groups [40].

We also found large losses at the stage of diagnosis, as approximately only one-third of all individuals with hypercholesterolemia was found to be aware of their high cholesterol. Our results further showed that age, high BMI, having diabetes, and having hypertension were significantly associated with being aware of one's high cholesterol level. This suggests that healthcare workers may appropriately prioritize those at greatest risk of CVD across the cascade, not only at the screening stage as described above. In the case of diabetes, this significant effect persisted through the final "control" stage of the cascade of care. This is an encouraging finding given the markedly worse CVD outcomes of patients with diabetes in comparison to those without [41]. Our results are in line with the current evidence base, as studies undertaken in several high- and upper-middle-income countries also found age and high CVD risk to be associated with greater awareness of having hypercholesterolemia [8,26,42]. Furthermore, while we did not find sex to be significantly associated with having received a hypercholesterolemia diagnosis, it is worth noting that prior studies have reported significant, albeit inconsistent patterns of sex differences [8,25,30].

The smallest loss in the care cascade—both in absolute and relative terms—occurred between the diagnosis and treatment stages. This is consistent with lifestyle advice essentially being cost free and previous evidence that found declining costs of cholesterol-lowering

medications in LMICs [8]. Nonetheless, a loss in care at this stage suggests that obstacles to treatment delivery persist. Here, previous evidence points toward a lack of access to and affordability of medicines, as well as the variation in treatment guidelines that influence clinical decisions [8,10,11,43]. On the international scale, this is reflected in the WHO's List of Essential Medicines, which currently includes only simvastatin for mixed hyperlipidemia. However, given the low treatment rates, expanding this list to include other statins, such as atorvastatin, pravastatin, fluvastatin, or lovastatin—which are currently only listed as therapeutic equivalents to simvastatin—could be one potential approach to increase their uptake [44].

Finally, a drop of 42% to 75% from all that received treatment for hypercholesterolemia to those that achieved control marked the largest relative loss in the care cascade. Both in the pooled analysis and at the country level, control rates were found to be low, ranging from virtually zero to 27% in all but 2 countries—Iran (57%) and Morocco (49%). This finding should be interpreted with the understanding that common treat-to-target ranges for lipids are not universally applied and are often combined with coronary heart disease risk levels, which could not be included in this study due to a lack of data availability [45]. Nonetheless, this finding is also reflected in other studies, where control rates in China, Thailand, and Jordan also ranged between 10% and 25% [8,30]. Such low control rates may reflect both insufficient treatment options available to providers, for instance, due to a lack of access to affordable medication as described above or due to poor treatment adherence by respondents. While improvements in medication availability may improve the former, a large literature base is currently forming around policy interventions such as mobile health or peer and community education to improve uptake and adherence to lipid-lowering therapy [46,47].

Generally, we found that the Americas, the Eastern Mediterranean, and European regions achieved higher cascade performance than Africa, Southeast Asia, and the Western Pacific regions. We further showed that upper-middle-income countries were consistently better at retaining individuals throughout the cascade than lower-middle income or low-income countries. This pattern may reflect that hypercholesterolemia care requires a level of attention that countries with low health system capacity may not be able or willing to achieve yet. Because hypercholesterolemia care is embedded in a framework of comprehensive CVD care, it is shaped by several clinical complexities of calculating risk scores, still comparably expensive screening and treatment options, and an international context that focuses on policies to target each of the cardiometabolic risk factors individually, for instance, through initiatives such as the WHO Global Diabetes Compact [10,11,33,48].

Within these patterns, we still found very large heterogeneity at the country level across all cascade stages, which is mirrored in the literature [8,25,26,30]. It is particularly noteworthy that Sri Lanka and Morocco were among the highest-performing countries—despite their lower-middle-income status. Sri Lanka has been shown to be highly engaged in fighting NCDs. They have a national NCD agenda, a high share of primary healthcare facilities that offer CVD risk management, cardiac rehabilitation programs, as well as policies targeting tobacco, alcohol and salt reductions, and NCDs generally [49–51]. Sri Lanka was also found to have the highest number of full-time equivalent professional staff in an NCD unit within the Ministry of Health in comparison to 6 other Asian countries [49]. The high performance of Morocco, on the other hand, was not mirrored in prior, yet limited literature. These studies have shown that while Morocco is already undergoing the epidemiological transition, the awareness of ischemic heart disease and CVD risk remained low in the population [52–54]. Hence, future research may yield valuable insights into the strengths and weaknesses of the Moroccan NCD care system. After Morocco and Sri Lanka, Costa Rica and Iran stood out as particularly high-performing, upper-middle-income countries. Notably, Costa Rica performed similarly well in corresponding analyses of the cascades of care for diabetes and hypertension,

which also further discuss potential reasons for its high performance [15,16]. In the cascade analysis for Iran, the high rates of controlled lipid values stood out in particular—which could be due to increasing statin prescriptions and food industry improvements and further speaks to Iran's high capacity for CVD control as well as its leading commitment in the Eastern Mediterranean region to fighting NCDs [52,55].

This study had several limitations. First, several measures may be subject to measurement biases. For one, our data on health services received were self-reported and thus may be subjected to a recall bias. For instance, individuals that were taking medication could have been more likely to remember ever being screened for hypercholesterolemia, affecting the absolute probability of reaching each cascade stage. Similarly, recalling the provision of low-touch treatment interventions, such as having received lifestyle advice, may be difficult for respondents. In addition, our definition of hypercholesterolemia was based on biomarkers that, in some countries, were measured with POC devices. While these may be less accurate than lab-based testing, studies have shown that they can be reliably used for lipid screening [29,56–58]. A study by Ferreira and colleagues (2015) found a 94.6% to 97.7% agreement between the CardioCheck PA—which is used by the majority of countries—and the laboratory when sorting lipid records into the ATP III lipid classifications used in our analysis [58]. Moreover, our disaggregated cascade analysis should be considered with the following caveats in mind. First, the comparability between countries is, to some extent, limited, as the time span of 9 years across surveys potentially introduced period effects into our analysis. While a cascade analysis by year showed no observable time trend (see S1L Fig), this must be viewed in light of the fact that the estimates are based on a small number of surveys per year and that they are likely heavily enmeshed with country effects. In addition, some country-level estimates have very small sample sizes due to low prevalence rates and are thus shown only for the purpose of completion. Relatedly, in our cascade regression analysis, we note that as we conditioned each cascade stage estimation on completion of prior cascade stages, increasing losses in sample size reduced the statistical power to detect significant associations—potentially explaining our findings. Finally, we chose to define hypercholesterolemia and achieving control based on the ATP III guidelines due to their frequent use in the literature. However, these are relatively conservative in comparison to some national guidelines, as is apparent when examining the markedly lower cascade performance when applying AHA/ACC guidelines (see S1C Fig). The comparability of countries is generally limited by a lack of universally used guidelines, as different guidelines are applied across countries and clinical settings and may even include geographical parameters, as is the case in the European Society of Cardiology (ESC) and European Atherosclerosis Society (EAS) guidelines [59,60]. Despite this, such a comparison still offers important insights into national and global care gaps and can be used for identifying effective policy as well as serve as markers of progress in tackling the burden of hypercholesterolemia.

## Conclusions

We found low levels of access to hypercholesterolemia care in this group of LMICs, with especially large levels of unmet screening and control needs across all countries. Further work is required to understand the underlying causes for this underperformance. A closer examination of the better performing countries in our study—such as Sri Lanka, Costa Rica, Iran, and Morocco—could yield important policy lessons, especially as the lipid cascade offers a potentially important tracer of unmet need for chronic disease care. Given its increasing relevance as one of the major, yet eminently preventable CVD risk factors, hypercholesterolemia deserves more attention both from a health services and a research perspective, globally.

## Supporting information

**S1 Text. Search methods.**
(DOCX)

**S2 Text. Country categories and country-specific sampling methods.**
(DOCX)

**S3 Text. Country-specific lipid measurement methods.**
(DOCX)

**S4 Text. Data cleaning.**
(DOCX)

**S5 Text. Mathematical equation to regression specifications.**
(DOCX)

**S6 Text. STROBE checklist.**
(DOCX)

**S1 Table. Sample characteristics.**
(DOCX)

**S2 Table. Supplementary analysis.**
(DOCX)

**S3 Table. Missing predictor variables by country among participants with hypercholester-olemia, by country.**
(DOCX)

**S1 Fig. Supplementary analysis.**
(DOCX)

## Acknowledgments

We would like to thank Clare Flanagan, Sarah Frank, Esther Lim, Yuanwei Xu, and Jacqueline Seiglie for help with data harmonization and translation of study documentation. We would also like to thank each of the country-level survey teams and study participants who made this analysis possible.

## Author Contributions

**Conceptualization:** Maja E. Marcus, Pascal Geldsetzer, Justine I. Davies, Mark A. Hlatky, Rifat Atun, Till W. Bärnighausen, Lindsay M. Jaacks, Jennifer Manne-Goehler, Sebastian Vollmer.

**Data curation:** Maja E. Marcus, Cara Ebert, Pascal Geldsetzer, Michaela Theilmann, Brice Wilfried Bicaba, Glennis Andall-Brereton, Pascal Bovet, Farshad Farzadfar, Mongal Singh Gurung, Corine Houehanou, Mohammad-Reza Malekpour, Joao S. Martins, Sahar Saeedi Moghaddam, Esmaeil Mohammadi, Bolormaa Norov, Sarah Quesnel-Crooks, Roy Wong-McClure, Rifat Atun, Till W. Bärnighausen, Lindsay M. Jaacks, Jennifer Manne-Goehler, Sebastian Vollmer.

**Formal analysis:** Maja E. Marcus.

**Investigation:** Maja E. Marcus, Jennifer Manne-Goehler, Sebastian Vollmer.

**Methodology:** Maja E. Marcus, Cara Ebert, Pascal Geldsetzer, Michaela Theilmann, Rifat Atun, Till W. Bärnighausen, Lindsay M. Jaacks, Jennifer Manne-Goehler, Sebastian Vollmer.

**Project administration:** Maja E. Marcus, Pascal Geldsetzer, Michaela Theilmann, Justine I. Davies, Rifat Atun, Till W. Bärnighausen, Lindsay M. Jaacks, Jennifer Manne-Goehler, Sebastian Vollmer.

**Resources:** Till W. Bärnighausen, Sebastian Vollmer.

**Software:** Maja E. Marcus, Pascal Geldsetzer, Michaela Theilmann, Sebastian Vollmer.

**Supervision:** Jennifer Manne-Goehler, Sebastian Vollmer.

**Validation:** Maja E. Marcus, Cara Ebert, Michaela Theilmann, Jennifer Manne-Goehler, Sebastian Vollmer.

**Writing – original draft:** Maja E. Marcus.

**Writing – review & editing:** Maja E. Marcus, Cara Ebert, Pascal Geldsetzer, Michaela Theilmann, Brice Wilfried Bicaba, Glennis Andall-Brereton, Pascal Bovet, Farshad Farzadfar, Mongal Singh Gurung, Corine Houehanou, Mohammad-Reza Malekpour, Joao S. Martins, Sahar Saeedi Moghaddam, Esmaeil Mohammadi, Bolormaa Norov, Sarah Quesnel-Crooks, Roy Wong-McClure, Justine I. Davies, Mark A. Hlatky, Rifat Atun, Till W. Bärnighausen, Lindsay M. Jaacks, Jennifer Manne-Goehler, Sebastian Vollmer.

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
