## [Editor Report · Decision Letter 0]

20 Apr 2021

Dear Dr Marcus, 

Thank you for submitting your manuscript entitled "Unmet Need for Dyslipidemia Care in Low- and Middle-Income Countries: A Cross-sectional Study of Nationally-representative Individual-level Data from 35 Countries" for consideration by PLOS Medicine.

Your manuscript has now been evaluated by the PLOS Medicine editorial staff and I am writing to let you know that we would like to send your submission out for external peer review.

Please re-submit your manuscript within two working days, i.e. by Apr 22 2021 11:59PM.

Kind regards,

Beryne Odeny

Associate Editor

PLOS Medicine

---

## [Decision Letter · Decision Letter 1]

20 May 2021

Dear Dr. Marcus,

Thank you very much for submitting your manuscript "Unmet Need for Dyslipidemia Care in Low- and Middle-Income Countries: A Cross-sectional Study of Nationally-representative Individual-level Data from 35 Countries" (PMEDICINE-D-21-01767R1) for consideration at PLOS Medicine. 

[LINK]

In light of these reviews, I am afraid that we will not be able to accept the manuscript for publication in the journal in its current form, but we would like to consider a revised version that addresses the reviewers' and editors' comments. Obviously we cannot make any decision about publication until we have seen the revised manuscript and your response, and we plan to seek re-review by one or more of the reviewers. 

We expect to receive your revised manuscript by Jun 10 2021 11:59PM. Please email us (plosmedicine@plos.org) if you have any questions or concerns.

We look forward to receiving your revised manuscript. 

Sincerely,

Beryne Odeny, 

PLOS Medicine

plosmedicine.org

1) Please revise your title according to PLOS Medicine's style. Your title must be nondeclarative and not a question. It should begin with main concept if possible. Please place the study design ("A cross-sectional study”) in the subtitle (ie, after a colon). For example, “Unmet Need for Dyslipidemia Care in 35 Low- and Middle-Income Countries: A Cross-sectional Study.“

2) In the abstract Methods and Findings:

a) Please structure your abstract using the PLOS Medicine headings (Background, Methods and Findings, Conclusions).

b) Please combine the Methods and Findings sections into one section, “Methods and findings”. 

c) Please ensure that all numbers presented in the abstract are present and identical to numbers presented in the main manuscript text.

d) Please quantify the main results (with 95% CIs and p values).

e) Please include the important dependent variables that are adjusted for in the analyses.

f) In the last sentence of the Abstract Methods and Findings section, please describe the main limitation(s) of the study's methodology.

3) Abstract summary - At this stage, we ask that you include a short, non-technical Author Summary of your research to make findings accessible to a wide audience that includes both scientists and non-scientists. The Author Summary should immediately follow the Abstract in your revised manuscript. This text is subject to editorial change and should be distinct from the scientific abstract. Please see our author guidelines for more information: https://journals.plos.org/plosmedicine/s/revising-your-manuscript#loc-author-summary.

4) Your study is observational and therefore causality cannot be inferred. Please remove language that implies causality, such as predictor/ predict. Refer to associations instead. 

5) Please add the following statement, or similar, to the Methods: "This study is reported as per the Strengthening the Reporting of Observational Studies in Epidemiology (STROBE) guideline (S1 Checklist)."

6) Please temper claims of primacy (e.g., “we are the first to apply) by stating, "to our knowledge" or something similar.

7) In the Methods and Results section:

a) Please provide 95% CIs and p values for all estimates.

b) When a p value is given, please specify the statistical test used to determine it.

c) Please present numerators and denominators for percentages, at least in the Tables

8) For your Tables and figures, please do the following:

a) Please indicate in the figure caption the meaning of the bars and whiskers

b) Please define the abbreviations such as BMI, TC, LDL-C, WHO PEN , PSU, ATP III

9) The terms gender and sex are not interchangeable (as discussed in http://www.who.int/gender/whatisgender/en/ ); please use the appropriate term.

10) Please replace "subject" with participant, patient, individual, or person.

11) Please replace “diabetics” with “persons with diabetes.” This applies also to similar terms like hypertensives

12) Please use the "Vancouver" style for reference formatting and see our website for other reference guidelines. For example, six names should appear before et al. https://journals.plos.org/plosmedicine/s/submission-guidelines#loc-references. Please ensure that weblinks are current and accessible to date.

13) Please include line numbers in the next revision

Comments from the reviewers:

Reviewer #1: The study examines the cascade of dyslipidemia care for people in 35 LMICs. I think this is an important study in that it fills the evidence gap of the unmet need of dyslipidemia care in LIMICs. 

Overall, this is a well-designed and carefully conducted study. However, I think there are still a few major issues that the authors need to address. 

First, although you noted that there are some systematic differences between the surveys (e.g., the level of blood lipids is measured differently across surveys), you assumed the data across surveys are comparable and did not adjust for the potential bias due to different sources. There are in fact approaches you could use to adjust this bias. An example could be the "bias correction" in this appendix of a Lancet study. 

(https://www.thelancet.com/cms/10.1016/S0140-6736(17)31833-0/attachment/d006d5eb-f74d-4fe3-a25f-2035410cfab3/mmc1.pdf)

Second, I believe it should be "modified Poisson regression" instead of "Poisson regression" that you used to examine the predictors of cascade progression. Poisson regression is mostly used for count outcomes while modified Poisson regression can be used on binary outcomes (https://academic.oup.com/aje/article/159/7/702/71883). However, I think modified Poisson is usually used for rare events and I don't think your outcomes are "rare". You might want to check the validity and justify the use of this regression for your analyses. Also, it would be better if you could provide a mathematical equation for the regression(s) in the paper. 

Lastly, if you chose modified Poisson because you wanted RR instead of OR, I personally think probability is a more intuitive measure for common readers. If you use logistic regression, which is more common than modified Poisson for binary outcomes, you can predict probability from the fitted logistic regressions and can compare the probability of cascade progression with different values of covariates. Here is a package you could use to do that. (https://faculty.washington.edu/cadolph/?page=60) 

Here are some specific comments about the manuscript. 

1. Please provide line numbers for the manuscript in the future. 

2. Page 6, Method/Data sources: The IHME's GHDx could be a great source for searching of survey data. 

3. Page 8, Method/ Cascade Construction: your definition of dyslipidemia includes people who were taking lipid-lowering medication. However, why did you exclude all "individuals without a biomarker record form the analysis"? What if these individuals do have record of taking lipid-lowering medication? It's not very clear here. 

4. Page 11, Results/ Sample Characteristics: "…Secondary schooling or higher education was completed by 41% of those with high TC and 31% of those with high LDL-C. Around 59-61% of those with dyslipidemia were overweight or obese and approximately 16-17% were current smokers…" Comparing the summaries with those among people without dyslipidemia could be more helpful.

5. Page 28, Table 2: There should be one overall p-value for the categorical variables.

Reviewer #2: Major Comments

I greatly appreciate the authors' work to put together this information for furthering global dyslipidemia care. This data was strikingly lacking and the current work contributes a step forward. I have few suggestions below and I hope the authors find this helpful. 

1. This sentence from the abstract is not clear. "The cascade analysis showed that 43% (95% CI: 40 to 45%) of subjects with high TC and 47% (95% CI: 44 to 50%) with high LDL-C had ever had their cholesterol measured." A corollary of this is - 57% of subjects with high TC and 53% with high LDL-C had not ever had their cholesterol measured. How is it possible for someone to have high TC or LDL-C, but not ever have cholesterol measured? 

2. Related to comment 1 is Figure 1 (and other similar figures): How is it possible to have dyslipidemia but lipids not measured? Is the difference solely from self-reported medication use? Even so, I can't imagine someone prescribing medication without checking lipids at some point. This actually makes many parts of the manuscript difficult to follow. Most likely, the respondents were not aware of lipids being measured previously. Please clarify this. Perhaps best approach would be remove the part "lipids measured" altogether from the cascade and change accordingly throughout the manuscript.

3. Has the cascade care methodology and performance been previously used in hypercholesterolemia or other chronic diseases? Can you please discuss something about its overall validity (at least, face validity) in the methods section? 

Minor Comments

1. "Normal" cholesterol level an evolving concept and is complex. For example, LDL-C of 130 as controlled dyslipidemia is fairly elevated from the current US, European and Canadian perspectives. Nevertheless, the cutoffs used in the current analysis is a conservative estimate, and should be acknowledged in the limitation section.

2. I would replace "dyslipidemia" on the title and throughout the manuscript by "hypercholesterolemia" since the latter is more specific to cholesterol than the former (dyslipidemia also implies markers beyond cholesterol, such as triglycerides). 

3. Cascade and its performance is a jargon and should be described at its first appearance in the abstract and manuscript. The terminology first appears in the results section of the abstract and the introduction section of the manuscript, but the description is seen only in the methods section of the manuscript.

4. I understand the difficulty in simplifying the complex data, but in Table 1 for the total cholesterol sample, why 7% of 128998 (=9029) is not the number of observations on the sample with high TC (N=10737)? Same comment for Table 4 in the appendix. Is it because the latter also includes people taking lipid lowering medications? If so, adding a clarifying sentence at the footnote will help the readers. Ditto question for LDL-C sample.

5. Figure 1 footnote reads: "Lipids Measured refers to the percentage share of all sick respondents (classified based on relevant biomarker) that have ever had their lipid status measured as per self-reported information." Sick respondents seem like the participants were severely ill, which is not necessarily the case in people with dyslipidemia. Please correct throughout the manuscript.

6. Not all advice given are remembered by patients, particularly lifestyle modification related advice. Thus, even though lifestyle changes were advised, participants may not remember them later. Please acknowledge.

7. Based on Table 3 similar proportion of people received medication and advice (Table 3, Appendix). However, it's unclear what proportion of people with high cholesterol only received advice without medications. This can be helpful information as unfortunately despite being generic, these medications may not be within reach of a large number of population. 

8. First the number of people with lipids measured in low income countries is small, and not surprisingly the cascade performance is poorest for low income countries (Table 5, appendix). Can the authors comment something about this in the discussion? The method of lipids measurement for 19 countries was based on point of care devices. One of the important barriers for dyslipidemia care worldwide is lack of access to lipid testing, which is needed more than once in many people. Perhaps using the POC device can fill this gap (within the constraints of its limitation). Making all statin lipid lowering therapy as an essential medication (of WHO) is a step in correct direction (currently only simvastatin is listed, which is one of the medication that we least likely to prescribe) and making them available in remote areas of all countries, including low income countries can partly address this gap. These elements could be discussed within the constructs of high performing countries. See last minor comment.

9. I am not following what the authors are saying in the text in the discussion section (underlined part). Please clarify or if of minor importance, remove. "This suggest that the targeting of care to people with high CVD risk, as described above for the awareness stage, also was preserved in later cascade stages - in the case of diabetes this included even the controlled stage. While recognizing that access to care remains low at the awareness stage, this is nevertheless an encouraging finding given the markedly worse CVD outcomes of patients with diabetes in comparison to those without." 

10. An average reader would have difficulty understanding this conclusion sentence. "We found poor cascade performance for dyslipidemia care in this group of LMICs, with large losses to care occurring at each stage of the cascade." I recommend rather describe what actually was seen - e.g., only a smaller percentage of those aware of diagnosis had cholesterol level controlled - something similar. 

11. I agree about highlighting the constructs of better performing countries, such as Sri Lanka for better cholesterol control. In addition to the conclusion, perhaps the authors can highlight this in their discussion section.

Reviewer #3: The present work is interesting and timely, since it affords the growth of dyslipidemias in Low and Middle Income Countries and the current unmet needs.

The study analyzes information from 35 Countries and is based on a sufficient sample size to draw conclusions (almost 130,000 cases). I have a few points to raise. The authors are invited to address them in revising the manuscript.

Most of my concerns, indeed, were already identified by the authors and listed in the "limitations" section.

However, I believe they deserve a more thorough discussion, mostly in view of the validity of the experimental approach and the interpretation of the results.

1. Dyslipidemia was assessed as a TC level exceeding 240 mg/dl and an LDL-C exceeding 160 mg/dl using ATP III. This represents today quite an obsolete way to define hypercholesterolemia and it should be replaced by a more contemporary, guideline-based definition, to avoid or minimize the risk of losing many subject which are indeed hypercholesterolemic and may deserve life style or pharmacologic interventions. (by the way, some of the Refs quoted in this regard 23-27 are not totally appropriate). Keeping in mind the latest widely accepted relationship between cholesterol and cv events (which should be mentioned), the thresholds selected in this analysis are well beyond levels considered unharmful

2. The source of information for lipid levels is extremely heterogeneous (self-reported, point-of-care, lab-based testing). It should be made an additional effort to provide a sub-analysis based on the different sources. This process, if guided by the authors, would most likely put the reader in a better shape than admitting that "our disaggregated cascade analysis should be followed with caveats in mind" (page 19), including also the very heterogeneous sample size due to specific country-limitation.

3. Discussion is very long and could be reduced in size. Despite that, it omits any comparison with high income countries, which in my view would be very informative and may add value to the study

4. Definition of Low-Middle income countries should be provided and supported by some informative data

5. Figures are nice. How about introducing one comparing with high-income countries? By the way data from the last Euro Aspire, for Instance, are not really satisfactory in terms of Cholesterol control. Achievements of satisfactory cholesterol control are still surprisingly low.

Reviewer #4: This is a large analysis of a combined dataset of 35 STEPS surveys from LMIC. It takes a rather cascade analytic approach to evaluate the proportion of adults with dyslipidemia who have received steps in care towards controlled disease. The strengths of the paper include: large dataset, multiple countries, and use of population-based cohorts (although unclear if all datasets population-based vs. population-representative).

Major limitations:

1. Major concern is this is supposed to reflect an international team reporting LMIC data and yet first, second, third, and last authors all not from the LMIC. This is not equitable in terms of modern data sharing, capacity building, and authorship. More, the interpretation of the data seems to lack a perspective from the field—from the LMIC perspective. The data reflects health systems gaps and the primary drop off in the cascade is "lipids measured"—meaning > 50% of those with dyslipidemia did not know they had it b/c they had not had lipids checked. This is likely b/c lipids are not available in their settings for many reasons! Yet authors make the major conclusions in their first paragraph of discussion that the findings are: gaps in treatment and control. 

2. Authors overstate the knowledge gap as give many references on dyslipidemia burden of disease including the GBD dataset. Appears principal innovation is applying the care cascade explicitly to data that is already available. 

3. Very unclear why authors choose to use adults > 15 years (with 40% of their sample < 35 years), but these are people who would not be expected to have lipids checked or treated with statins?

4. Do not report how many ppts are excluded from the dataset who did not have lipids measured. Is Table 1 denominator truly 100% of each country's STEPS cohort? 

5. Do not address how representative their pooled dataset is compared to all LMICs. While they have 35 countries, many countries are quite small population wise and may only make up a small proportion of global burden of dyslipidemia

6. Do not address missing data in general

7. Discussion is unfocused --see detailed comments in the attached full review . The main points appear to be ~7% prevalence and largest drop off is lipid measurement--yet discussion organized as main points are treatmetn and control so seems mis-aligned. Moreover, the discussion rambles to report other papers cascade % without any nuance of explaining or hypothesizing why. 

Additional comments in detail in the attached.

[LINK]

---

## [Decision Letter · Decision Letter 2]

22 Sep 2021

Dear Dr. Marcus,

Thank you very much for re-submitting your manuscript "Unmet Need for Hypercholesterolemia Care in 35 Low- and Middle-Income Countries:

A Cross-sectional Study of Nationally-representative Surveys" (PMEDICINE-D-21-01767R2) for review by PLOS Medicine.

I have discussed the paper with my colleagues and the academic editor and it was also seen again by four reviewers. I am pleased to say that provided the remaining editorial and production issues are dealt with we are planning to accept the paper for publication in the journal.

[LINK]

We look forward to receiving the revised manuscript by Sep 29 2021 11:59PM.   

Sincerely,

Beryne Odeny, 

Associated Editor 

PLOS Medicine

plosmedicine.org

Requests from Editors:

1) In the abstract:

a) Please remove the limitations from the Conclusion and include it as the last sentence of the abstract’s Methods and Findings section.

2) Please title the summary you have provided after the abstract as "Author summary." Please see our author guidelines for more information: https://journals.plos.org/plosmedicine/s/revising-your-manuscript#loc-authorsummary. 

3) In the main text, please provide p-values in addition to 95% CI where appropriate.

4) Please provide the total N in tables 1 and 2

5) For your Tables and figures, please do the following:

a) Please indicate in the figure caption the meaning of the bars and whiskers

b) Please define the abbreviations such as BMI, TC, LDL-C, WHO PEN , PSU, ATP III

Comments from Reviewers:

Reviewer #1: The authors have made great efforts to address the reviewers' comments and I appreciate that. I think the authors have addressed my comments and other reviewers' comments very well. The responses are respectful, clear and in details. 

I only have one follow-up comment on presenting probability rather than RR or OR for the regression model. I think you misunderstood my suggestion. I did not ask for a linear probability model. What I am proposing is that even if you fitted the data using modified poisson (or logistic) regression, you can always present the effect of an intervention in term of change in the probability of the outcome instead of RR or OR by using the coefficients from the fitted model and some simulation techniques. I think probability is much more intuitive to most non-academic people than RR or OR is. 

That being said, I understand the reason why you used modified Poisson and I agree that presenting RR is common and is fine.

Reviewer #2: I appreciate the authors' work on the revision. I only have two minor comments. Defining "LDL-C measurement of 70 mg/dL or higher as having hypercholesterolemia" is very aggressive (as this is not unanimously accepted definition in all patients even with the current ACC/AHA guidelines), but I suggest phrasing this analysis as assessing how results vary across the spectrum. 

On page 10: "TC was smaller than 200 mg/dL and LDL-C was smaller than 122 130 mg/dL." I assume you meant lower than and not smaller than. 

Reviewer #3: The authors have addressed all comments of this Reviewer.

Reviewer #4: Authors respond the vast majority of suggested revisions adequately and paper is strengthened including additional analyses and clearer description of limitations and inference.

[LINK]

---

## [Editor Report · Decision Letter 3]

8 Oct 2021

Dear Dr Marcus, 

On behalf of my colleagues and the Academic Editor, Dr. Aaron S Kesselheim, I am pleased to inform you that we have agreed to publish your manuscript "Unmet Need for Hypercholesterolemia Care in 35 Low- and Middle-Income Countries:

A Cross-sectional Study of Nationally-representative Surveys" (PMEDICINE-D-21-01767R3) in PLOS Medicine.

PRESS

Sincerely, 

Beryne Odeny 

PLOS Medicine